# Exosomal MALAT1 from Rapid Electrical Stimulation-Treated Atrial Fibroblasts Enhances Sox-6 Expression by Downregulating miR-499a-5p

**DOI:** 10.3390/cells13231942

**Published:** 2024-11-22

**Authors:** Cheng-Yen Chuang, Bao-Wei Wang, Ying-Ju Yu, Wei-Jen Fang, Chiu-Mei Lin, Kou-Gi Shyu, Su-Kiat Chua

**Affiliations:** 1Division of Cardiology, Department of Internal Medicine, Shin Kong Wu Ho-Su Memorial Hospital, Taipei 11101, Taiwan; m014190@ms.skh.org.tw (C.-Y.C.); baowei@ms22.hinet.net (B.-W.W.); kinki1983yu@gmail.com (Y.-J.Y.); wjfang0719@gmail.com (W.-J.F.); mei882153@gmail.com (C.-M.L.); shyukg@ms12.hinet.net (K.-G.S.); 2Department of Emergency Medicine, Shin Kong Wu Ho-Su Memorial Hospital, Taipei 11101, Taiwan; 3School of Medicine, College of Medicine, Fu Jen Catholic University, New Taipei 24205, Taiwan

**Keywords:** microRNA-499a-5p, MALAT1, sox-6, rapid electrical stimulation, apoptosis

## Abstract

Background: Atrial fibrillation (AF) is a common cardiac arrhythmia associated with significant morbidity and mortality. Rapid electrical stimulation (RES) of atrial fibroblasts plays a crucial role in AF pathogenesis, but the underlying molecular mechanisms remain unclear. This study investigates the regulatory axis involving MALAT1, miR-499a-5p, and SOX6 in human cardiac fibroblasts from adult atria (HCF-aa) under RES conditions. Methods: HCF-aa were subjected to RES at 0.5 V/cm and 10 Hz. The expression levels of metastasis-associated lung adenocarcinoma transcript 1 (MALAT1), miR-499a-5p, and SRY-Box Transcription Factor 6 (SOX6) were measured using qPCR and Western blot analyses. Luciferase reporter assays were performed to confirm target relationships. The effects of MALAT1 siRNA, miR-499a-5p mimics/inhibitors, and SOX6 overexpression on gene expression and apoptosis were assessed. Results: RES increased exosomal MALAT1 expression, peaking at 2 h. MiR-499a-5p levels initially increased, then decreased at 2 h, coinciding with peak MALAT1 expression. SOX6 mRNA and protein levels increased, peaking at 4 and 6 h, respectively. Luciferase assays confirmed MALAT1 and SOX6 as miR-499a-5p targets. MALAT1 knockdown increased miR-499a-5p levels and reduced SOX6 expression. MiR-499a-5p overexpression decreased SOX6 levels and inhibited RES-induced apoptosis. Conclusion: In HCF-aa under RES, increased exosomal MALAT1 expression counteracts miR-499-5p’s suppression of SOX6, suggesting that MALAT1-containing exsosomes derived from HCF-aa may offer a novel cell-free therapeutic approach for AF.

## 1. Introduction

Atrial fibrillation (AF) is the most common sustained cardiac arrhythmia encountered in clinical practice, affecting over 33 million individuals worldwide [1]. The presence of AF is associated with an increased risk of stroke, heart failure, and overall mortality, thereby posing a significant burden on healthcare systems [2]. Emerging evidence suggests that the rapid electrical stimulation (RES) of atrial fibroblasts plays a crucial role in the pathogenesis of AF [3]. Fibroblasts, traditionally thought to be responsible for maintaining structural integrity, have recently been recognized as active participants in the development and perpetuation of AF through a process known as fibroblast–myocyte coupling [4].

The cellular and molecular mechanisms that underpin the role of fibroblasts in AF pathophysiology remain a mystery. Recent studies, however, have shed light on a novel aspect of the involvement of long non-coding RNAs (lncRNAs) and microRNAs (miRNAs) as crucial regulators of gene expression in various cardiovascular diseases, including AF [4,5]. LncRNAs, which are transcripts longer than 200 nucleotides and lack protein-coding potential, can modulate gene expression through diverse mechanisms, such as acting as molecular sponges for miRNAs [6]. In contrast, miRNAs, small non-coding RNAs, post-transcriptionally regulate gene expression by binding to the 3′-untranslated region (3′-UTR) of target messenger RNAs (mRNAs), leading to translational repression or mRNA degradation.

Metastasis-associated lung adenocarcinoma transcript 1 (MALAT1) is a highly conserved lncRNA implicated in various physiological and pathological processes, including cancer, cardiovascular diseases, and neurological disorders [5]. MALAT1 acts as a competing endogenous RNA (ceRNA) by binding to and sequestering miRNAs, thereby preventing them from interacting with their target mRNAs [5]. Notably, MALAT1 has been shown to regulate the expression of several miRNAs, such as miR-499a-5p, which plays a crucial role in cardiac development, homeostasis, and disease [6].

MiR-499a-5p is a muscle-specific miRNA encoded within the intron of the MYH7B gene, which encodes the β-myosin heavy chain [7]. Previous studies have demonstrated that miR-499a-5p is involved in various cardiac processes, including myocyte differentiation, hypertrophy, and apoptosis [8]. Furthermore, miR-499a-5p has been implicated in regulating SOX6 (SRY-Box Transcription Factor 6), a transcription factor critical in cardiac development and disease [9]. SOX6 is a member of the SOX family of transcription factors characterized by their ability to bind to the consensus DNA sequence AACAAT [10]. Additionally, SOX6 has been implicated in regulating cardiac fibrosis, a hallmark of AF pathogenesis [9].

In the context of AF, it has been suggested that RES of atrial fibroblasts may lead to alterations in the expression of lncRNAs, miRNAs, and their target genes, ultimately contributing to the development and perpetuation of the arrhythmia [11]. However, the precise mechanisms by which RES modulates the interplay between lncRNAs, miRNAs, and their target genes in atrial fibroblasts remain poorly understood.

This study aims to elucidate the molecular mechanisms underlying the regulation of MALAT1, miR-499a-5p, and SOX6 in human cardiac fibroblasts derived from adult atria (HCF-aa) subjected to RES. Specifically, we hypothesize that RES induces the expression of MALAT1, which sequesters miR-499a-5p, leading to the upregulation of SOX6. We further postulate that this regulatory axis plays a crucial role in the pathogenesis of AF by modulating fibroblast function and contributing to the development of fibrotic remodeling.

## 2. Materials and Methods

### 2.1. Cell Culture of Human Cardiac Fibroblasts-Adult Atrial

Human cardiac fibroblasts from adult atria (HCF-aa) were obtained from ScienCell Research Laboratories (San Diego, CA, USA). These cells were cultured in a specialized fibroblast medium enriched with essential and nonessential amino acids, vitamins, organic and inorganic compounds, hormones, growth factors, and trace minerals. The medium contained a low concentration of fetal bovine serum (2%) and was buffered with HEPES and bicarbonate to maintain a stable pH of 7.4. The cells were incubated in a humidified atmosphere with 5% CO_2_ at 37 °C to ensure optimal growth conditions.

### 2.2. In Vitro Rapid Electrical Stimulation of Cultured HCF-aa Cells

HCF-aa cells were subjected to RES using a previously established protocol [3]. Once the HCF-aa cells reached confluence, the culture medium was replaced with serum-free Fibroblast Medium (ScienCell, Cat No: 2301). Following six hours of serum-free incubation, the cells were transferred to an EPS-01 culture pacer dish (BioEast, Taipei, Taiwan) for electrical stimulation (refer to Appendix A).

The electrical stimulation was administered using a RES generator inside a 5% CO_2_ incubator maintained at 37 °C. A custom-built stimulator with two parallel platinum electrodes was used to prevent electrolysis. This setup delivered a biphasic square waveform at a stimulation frequency of 10 Hz, with each pulse having a duration of 50 ms and an interval of 50 ms between pulses. The voltage gradient applied was 0.5 V/cm, which was determined from preliminary tests showing that the capture threshold for HCF-aa cells ranged from 0.25 to 0.75 V/cm.

Cell samples were collected at 0.5, 1, 2, 4, 6, and 8 h post-stimulation. For comparison, a control group of HCF-aa cells was maintained under identical conditions but without electrical stimulation.

### 2.3. Extraction of Exosomes from Cell Media

Exosomes were extracted from the cell culture media using the Total Exosome Isolation Reagent from Invitrogen, Thermo Fisher Scientific, adhering to the manufacturer’s instructions. This extraction process followed well-established protocols [12,13]. The quantification of the exosomes was performed using the ExoQuant™ quantification assay kit, according to the guidelines provided by BioVision (Milpitas, CA, USA).

### 2.4. Reverse Transcription and Real-Time Quantitative Polymerase Chain Reaction

Total RNA was isolated from the cytosol using the TRIzol reagent (Invitrogen, Thermo Fisher Scientific, Waltham, MA, USA), following the manufacturer’s instructions. For reverse transcription, 2 μg of total RNA was converted into cDNA using random hexamers and the High-Capacity cDNA Reverse Transcription Kit (Applied Biosystems, Thermo Fisher Scientific, Waltham, MA, USA).

Quantitative Polymerase Chain Reaction (PCR) used 10% of the cDNA as the template. The reactions were prepared with Fast SYBR Green Master Mix (Applied Biosystems) and run on an ABI StepOnePlus cycler under the following conditions: an initial denaturation at 95 °C for 15 min, followed by 40 cycles of 94 °C for 15 s, 55 °C for 30 s, and 70 °C for 30 s. Gene expression levels were quantified using the 2^–ΔCT^ method, where ΔCT = CT(target gene) − CT(control). Product specificity and purity were confirmed through sequence analysis.

Exosomes and cytosol were processed using the mirVana miRNA Isolation Kit (Invitrogen) for miRNA analysis. Reverse transcription of miRNA was performed using 5 μL of the isolated miRNA and the TaqMan miRNA Reverse Transcription Kit (Applied Biosystems). The qPCR analysis for miRNAs was conducted on 10% of the cDNA using TaqMan Universal PCR Master Mix (Applied Biosystems) with an ABI StepOnePlus cycler. The thermal cycling conditions were as follows: 50 °C for 2 min, 95 °C for 10 min, and then 40 cycles of 95 °C for 15 s and 60 °C for 1 min. Relative miRNA expression was calculated using the 2^–ΔCT^ method, where ΔCT = CT(target miRNA) − CT(control). The purity of the PCR products was validated through sequence analysis.

### 2.5. Construction and Delivery of the miR499 Antagomirs and Mutant microRNA Expression Vectors into Cultured HCF-aa

The procedure for generating the 85bp miR-499a precursor construct began with the amplification of genomic DNA using the forward primer CACGCCCTCTGCAGGC and the reverse primer CAGGACTCCCTCCCATGG. This process produced a 200 bp amplified product, which was digested with EcoRI and BamHI restriction enzymes.

In the mutant, the miR-499a precursor sequence was altered from CATTATTACTTTTGGTACGCG to CTAATAAAGTTTTGGTAGGCG. In the antagomiR, it was mutated to AAACATCACTGCAAGTCTTAA. Constructs were sequence verified. Following this, the miR-499 antagomir and a mutant miR-499 precursor (Applied Biosystems) were similarly designed and ligated into the same pmR-ZsGreen1 plasmid vector as the miR-499 construct. The constructed plasmids (2 μg) were transfected by ViaFect™ Transfection Reagent (Promega, Madison, WI, USA) in accordance with the manufacturer’s protocol. Briefly, incubating the mixture of transfection reagent and plasmid DNA for 20 min at room temperature was performed. We then added the mixture to culture cell medium and incubated cells at 37 °C for 24 h.

### 2.6. Transfection of MALAT1 Locked Nucleic Acid (LNA) GapmeR

Cells were transfected at 60% to 75% confluence with 5nM synthesized small interfering RNA (siRNA) targeting MALAT1 (Cat.no.4392420, Thermo Fisher Scientific, Waltham, MA, USA) using Lipofectamine RNAiMax (Thermo Fisher Scientific, Waltham, MA, USA) according to the manufacturer’s protocol. Control siRNA (sc-37007, Santa Cruz Biotechnology, Paso Robles, CA, USA) was transfected as negative control.

### 2.7. Luciferase Activity Assay

A luciferase reporter assay was introduced to verify the relationship between miR-499a-5p and MALAT1 and between miR-499a-5p and Sox-6-3′UTR, in the HCF-aa under RES.

A 500 bp human SOX-6-3′UTR DNA fragment was generated (Chromosome 11: 15,966,449-16,402,867; http://www.ensembl.org/index.html), accessed on 15 April 2020, and a human MALAT1 DNA fragment was generated (Chromosome 11: 65,497,762–65,506,469; http://www.ensembl.org/index.html), accessed on 15 April 2020, by artificial synthesis and cloning them into the pUC57 vector. The clone was digested with Sac I and Xba I restriction enzymes and ligated into a pmirNanoGLO luciferase plasmid vector. The human SOX-6-3′UTR contained miR-499a-5p conserved sites at SOX-6-3′UTR (from 609 to 629 bp). For the mutant, the conserved sites, AGGAGACACTGCAAAGTCTTAG, were mutated into CGGAGAACAGTACCATGAGGCG and was constructed by the same aforementioned method. The human MALAT1 contained miR-499a-5p conserved sites at MALAT1 (from 839 to 859 bp). For the mutant, the conserved sites, AACCGTCCCTGCAAGGCTGGG, were mutated into CCCAGGACAGTACCTGAGGGG and were constructed by the same aforementioned method. All the cloned plasmids were confirmed by DNA sequencing (Seeing Bioscience Co. Ltd., Taipei, Taiwan). The constructed plasmids (2 μg) were transfected by ViaFect™ Transfection Reagent (Promega) in accordance with the manufacturer’s protocol. Briefly, incubating the mixture of transfection reagent and plasmid DNA for 20 min at room temperature was performed. We then added the mixture to the culture cell medium and incubated cells at 37 °C for 24 h. Following treatment, cells extraction were prepared using the Nano-Glo dual-luciferase reporter assay system (Promega) and measured for luciferase activity by using a luminometer (Glomax Multi Detection System, Promega).

### 2.8. Western Blot Analysis

Cells were harvested by scraping and centrifuged at 300× *g* for 10 min at 4 °C. The resulting cell pellet was re-suspended and homogenized in a lysis buffer (Promega, Madison, WI, USA). The homogenate was then centrifuged at 10,600× *g* for 20 min. Protein concentration was determined using the Bio-Rad Protein Assay.

A Bio-Rad Protein Assay was used to measure protein content. Equal amounts of protein (30 μg) were loaded onto 10% sodium dodecyl sulfate-polyacrylamide gels and subjected to electrophoresis. The subsequent steps of the Western blot analysis were performed as previously described [13].

### 2.9. Flow Cytometric Analysis for Quantifying Apoptosis

Apoptotic cells were identified and quantified using flow cytometry by determining the percentage of cells with hypodiploid (sub-G1) DNA content. HCF-aa were fixed with 70% ethanol, treated with RNase to remove RNA, and then stained with propidium iodide (50 μg/mL, Molecular Probes, Eugene, OR, USA) and fluorescein isothiocyanate-conjugated annexin V (FITC Annexin V Apoptosis Detection Kit 1, Cat No.: 556547, BD Pharmingen, Franklin Lakes, NJ, USA). The HCF-aa population was categorized into three groups based on their staining patterns: cells negative for both annexin V and propidium iodide were considered alive, cells positive for annexin V but negative for propidium iodide were considered to be undergoing early-stage apoptosis, and cells positive for both annexin V and propidium iodide were classified as being in the late stage of apoptosis, also known as secondary apoptosis. A FACSCalibur flow cytometer equipped with Cell Quest software (https://www.bdbiosciences.com, Becton Dickinson, Franklin Lakes, NJ, USA) was used to measure DNA content and analyze the staining patterns, with 10,000 cells counted in each assay to ensure statistical significance.

### 2.10. Statistical Analysis

Data are presented as mean ± standard error of the mean (SEM). Statistical analyses were conducted using the GraphPad Software platform (https://www.graphpad.com/, San Diego, CA, USA). A one-way analysis of variance (ANOVA) was employed to assess statistical significance among multiple groups. Dunnett’s test compared multiple groups to a single control group. The Tukey–Kramer method was applied for pairwise comparisons between multiple groups following ANOVA. Statistical significance was set at a *p*-value of less than 0.05.

## 3. Results

### 3.1. Effect of Rapid Electrical Stimulation on the Expression of MALAT1, miR-499a-5p, and Sox-6 in Cultured HCF-aa

Real-time quantitative PCR was used to measure mRNA levels to investigate the impact of RES on MALAT1, miR-499a-5p, and Sox-6 expression in HCF-aa. RES at 0.5 V/cm and 10 Hz significantly increased exosomal MALAT1 mRNA expression compared to the control, 0.25 V/cm, and 0.75 V/cm conditions (Figure 1A). Based on these findings, these specific culture conditions were selected, and different durations of RES were tested to assess the expression of MALAT1, miR-499a-5p, and Sox-6 in cultured HCF-aa.

Following RES at 0.5 V/cm and 10 Hz, the exosomal MALAT1 levels in HCF-aa increased gradually, peaking at 2 h of RES and decreasing gradually, reaching the lowest point at 8 h (Figure 1B). Under these different conditions, the levels of cytoplasmic MALAT1 did not show significant changes (Figure 1C).

Cytoplasmic miR-499a-5p showed significant upregulation at 0.5 h of RES. After 1 h, its levels began to decrease, reaching the lowest point at 2 h. However, in subsequent measurements at 4, 6, and 8 h, miR-499a-5p levels gradually increased (Figure 2A). Pretreatment with MALAT1 siRNA increased the level of miR-499a-5p in HCF-aa at 2 h of RES. Additional control siRNA did not affect miR-499a-5p expression under the same conditions (Figure 2B).

In contrast, cytoplasmic Sox-6 mRNA exhibited significant upregulation at 1 h of RES and continued to rise, peaking at 4 h (Figure 3A). Similarly, the level of Sox-6 protein began to increase after 1 h, reaching its highest point at 6 h of RES (Figure 3B).

### 3.2. MALAT1 and Sox-6 Are Target Genes of miR-499a-5p in HCF-aa Under RES

To investigate whether MALAT1 and Sox-6 are target genes of miR-499a-5p in HCF-aa under RES, MALAT1-3′-UTR, and Sox-6-3′-UTR luciferase activity assays were performed. Figure 4A shows the sequence of the MALAT1 3′UTR target for miR-499a-5p binding (nucleotides 839–859). Pretreatment with antagomir-499a-5p significantly upregulated wild-type MALAT1 expression compared to HCF-aa under RES alone. The inhibition of MALAT1-3′-UTR luciferase activity expression was attenuated in HCF-aa under RES (0.5 V/cm, 10 Hz) for 0.5 h if its binding site for miR-499a-5p was mutated (Figure 4B). These findings indicate that MALAT1 is a target gene of miR-499a-5p.

We discovered that the Sox-6 3′UTR (nucleotides 609–629) had a binding site for miR-499a-5p, as indicated in Figure 4C. Pretreatment with antagomir-499a-5p significantly upregulated wild-type Sox-6-3′UTR luciferase activity in HCF-aa under RES for 0.5 h. The inhibitory effect of RES (0.5 V/cm, 10 Hz) on Sox-6-3′UTR expression for 0.5 h was attenuated when the binding site for miR-499a-5p on Sox-6-3′UTR was mutated (Figure 4D). These findings provide strong evidence that Sox-6 is indeed a target gene of miR-499a-5p.

### 3.3. MALAT1 Regulates Sox-6 Expression via miR-499a-5p in HCF-aa Under Rapid Electrical Stimulation

This study examined the molecular regulation of miR-499a-5p, MALAT1, and Sox-6 in HCF-aa under RES, as depicted in Figure 5. RES (0.5 V/cm, 10 Hz) treatment for 4 and 6 h significantly upregulated Sox-6 mRNA and protein expression in HCF-aa compared to the control group. The overexpression of wild-type miR-499a-5p reduced Sox-6 mRNA and protein expression. At the same time, pretreatment with antagomiR-499a-5p and mutant miR-499a-5p upregulated Sox-6 mRNA and protein expression in a similar pattern in HCF-aa under RES for 4 and 6 h, respectively. Additional MALAT1 siRNA attenuated the expression of Sox-6 mRNA and protein in HCF-aa under RES for 4 and 6 h, respectively. These findings suggest that MALAT1 downregulates miR-499a-5p and consequently upregulates Sox-6 expression in HCF-aa under RES. Scramble siRNA had minimal impact on Sox-6 expression in HCF-aa under RES.

### 3.4. RES-Induced Apoptosis Was Mediated by miR-499a-5p in HCF-aa

Propidium iodide–annexin V double-staining and FACS analysis were used to evaluate apoptosis. The percentage of HCF-aa apoptosis after RES (0.5 V/cm, 10 Hz) for 2 h was significantly higher than in the control group (Figure 6A,B). The increased propidium iodide and annexin V staining were reversed considerably by miR-499a-5p overexpression (Figure 6C), whereas mutant miR-499a-5p did not affect apoptosis (Figure 6D). Pretreatment with MALAT1 siRNA inhibited apoptosis after RES for 2 h (Figure 6E), whereas the control siRNA had no effect in HCF under RES for 2 h (Figure 6F). Figure 6G shows the apoptotic quantification results of cells stained with both annexin V and propidium iodide. These results indicate that miR-499a-5p and MALAT1 are essential in HCF-aa apoptosis after RES.

## 4. Discussion

This study provides novel insights into the molecular mechanisms underlying HCF-aa’s response to RES, with a focus on the interplay between the long non-coding RNA MALAT1, microRNA-499a-5p (miR-499a-5p), and the transcription factor SOX6. Our findings suggest a complex regulatory network that may contribute to the pathogenesis of atrial fibrillation (AF).

RES significantly increased exosomal MALAT1 expression in HCF-aa, peaking at 2 h post-stimulation. This finding aligns with previous studies demonstrating that MALAT1 is responsive to various cellular stresses, including hypoxia and oxidative stress [14]. The rapid upregulation of exosomal MALAT1 suggests it may serve as an early mediator of the cellular response to electrical stimulation. Interestingly, cytoplasmic MALAT1 levels remained relatively stable, indicating a specific mechanism for exosomal packaging and secretion of this lncRNA in response to RES. 

The dynamic changes in miR-499a-5p expression following RES are particularly intriguing. The initial upregulation at 0.5 h, followed by a significant decrease at 2 h, coincides with the peak of exosomal MALAT1 expression. This inverse relationship supports our hypothesis that MALAT1 may act as a molecular sponge for miR-499a-5p, as demonstrated by our luciferase assay results. The gradual increase in miR-499a-5p levels after 4 h suggests a complex regulatory mechanism, possibly involving feedback loops or compensatory responses [15]. 

Our findings regarding SOX6 expression further support the proposed regulatory axis. The upregulation of SOX6 mRNA and protein levels following RES, peaking at 4 and 6 h, respectively, is consistent with the downregulation of miR-499a-5p. This temporal relationship, combined with our luciferase assay results, provides strong evidence that SOX6 is a target of miR-499a-5p in HCF-aa under RES conditions. These findings align with previous studies identifying SOX6 as a target of miR-499 in other cellular contexts [16].

MALAT1 siRNA and miR-499a-5p antagomirs/mimics provided further insights into the regulatory relationships between these molecules. The observation that MALAT1 knockdown increases miR-499a-5p levels and subsequently reduces SOX6 expression supports the proposed mechanism of MALAT1 acting as a ceRNA for miR-499a-5p. This is consistent with the growing evidence suggesting that lncRNAs can modulate miRNA function through sequestration [17]. 

The impact of this regulatory axis on HCF-aa apoptosis is particularly relevant to AF pathogenesis. Our flow cytometry results demonstrate that RES-induced apoptosis is mediated, at least in part, by miR-499a-5p. The ability of MALAT1 siRNA to inhibit apoptosis further underscores the importance of this regulatory network in determining cellular fate under stress conditions. These findings align with previous studies implicating both MALAT1 and miR-499 in regulating cardiomyocyte apoptosis [18].

The observed effects on apoptosis may have significant implications for AF pathogenesis. Fibroblast apoptosis has been shown to contribute to atrial fibrosis, a key feature of AF [19]. By modulating fibroblast survival, the MALAT1/miR-499a-5p/SOX6 axis may influence the extent of fibrotic remodeling in the atria, potentially affecting the propensity for AF development or persistence.

Furthermore, the rapid induction of exosomal MALAT1 by RES suggests a potential mechanism for intercellular communication in the context of AF. Exosomes have been increasingly recognized as important mediators of cell-to-cell signaling in various cardiovascular diseases [20]. The packaging and secretion of MALAT1 in exosomes may allow for transmitting stress signals to neighboring cells, potentially amplifying the response to electrical stimulation across the atrial tissue.

The involvement of SOX6 in this regulatory network is particularly intriguing, given its known roles in cardiac development and disease. SOX6 has been implicated in regulating cardiac gene expression and has been shown to play a role in maintaining the identity of cardiomyocytes [21]. Our findings suggest that SOX6 may also have essential functions in adult cardiac fibroblasts, potentially influencing their phenotype or behavior under stress conditions.

While our study provides valuable insights into the molecular mechanisms underlying the response of atrial fibroblasts to RES, several limitations should be acknowledged. Firstly, our experiments were conducted in an in vitro system, which may not fully recapitulate the complex in vivo environment of the atria. Future studies using animal models of AF would be valuable in confirming the relevance of this regulatory axis in vivo. Secondly, while we have focused on the MALAT1/miR-499a-5p/SOX6 axis, it is likely that other lncRNAs, miRNAs, and transcription factors are also involved in the response to RES. A more comprehensive analysis of the transcriptome and proteome changes induced by RES would provide a fuller picture of the molecular events underlying AF pathogenesis.

## 5. Conclusions

In conclusion, our study reveals a novel regulatory axis involving MALAT1, miR-499a-5p, and SOX6 in the response of atrial fibroblasts to rapid electrical stimulation (Figure 7). This mechanism may contribute to the pathogenesis of AF by modulating fibroblast survival and potentially influencing fibrotic remodeling. These findings advance our understanding of the molecular basis of AF and suggest potential new targets for therapeutic intervention. Future studies should aim to elucidate further this regulatory axis’s downstream effects on fibroblast function and explore its potential as a target for AF prevention and treatment.

## Figures and Tables

**Figure 1 cells-13-01942-f001:**
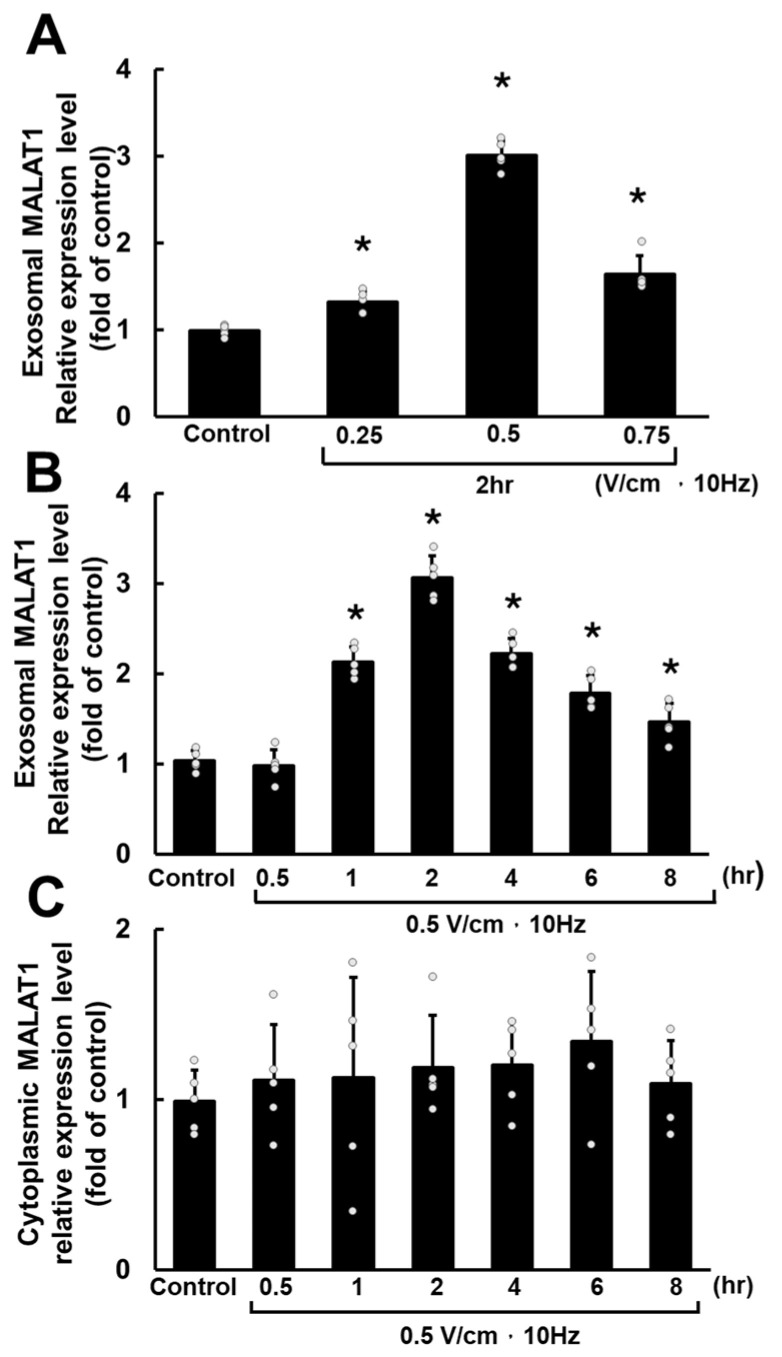
Rapid electrical stimulation (RES) affects exosomal and cytoplasmic MALAT1 expression in HCF-aa. (**A**) MALAT1 mRNA expression in HCF-aa subjected to RES at varying voltages (10 Hz for 2 h). The control group was cultured for 2 h without RES. (**B**) Time course of exosomal MALAT1 mRNA expression in HCF-aa subjected to RES (0.5 to 8 h). (**C**) Time course of cytoplasmic MALAT1 mRNA expression in HCF-aa subjected to RES (0.5 to 8 h). (**B**,**C**) The control group was cultured for 8 h without RES. Data represent mean ± SEM from three independent experiments (n = 5 per group). * *p* < 0.001 vs. control group.

**Figure 2 cells-13-01942-f002:**
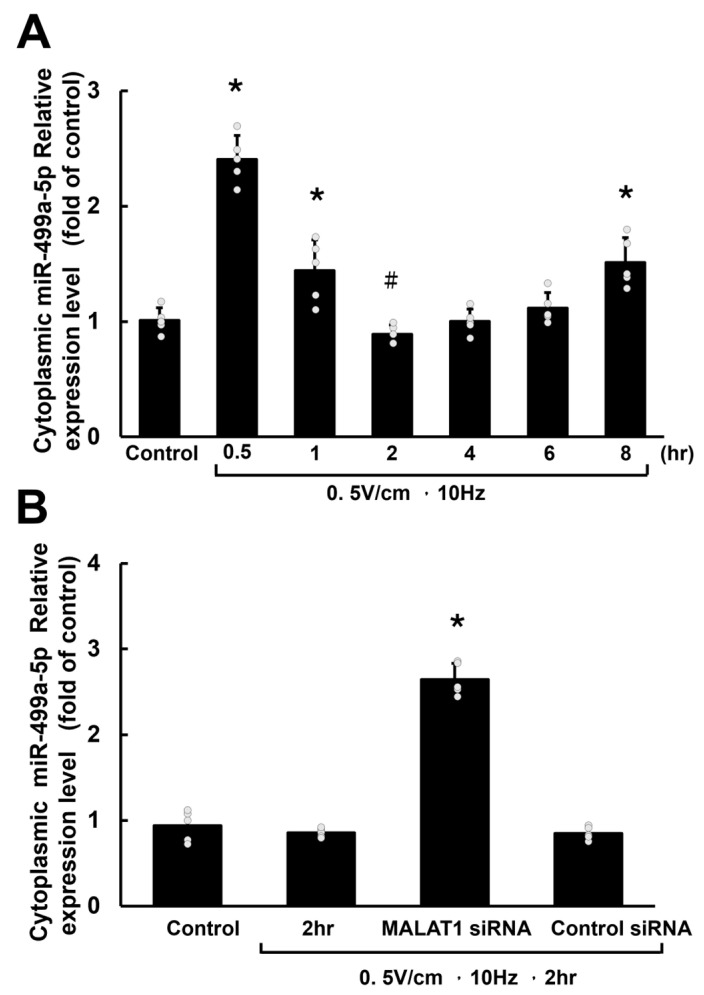
Cytoplasmic miR-499-5p expression in HCF-aa under RES. (**A**) Time course of relative cytoplasmic miR-499a-5p expression levels in HCF-aa subjected to RES for 0.5 to 8 h. The control group was cultured for 8 h without RES. (**B**) Representative cytoplasmic miR-499a-5p expression under various experimental conditions in HCF-aa subjected to RES. Data represent mean ± SEM from three independent experiments (n = 5 per group). * *p* < 0.001 vs. control group; # *p* < 0.001 vs. RES group for 0.5 hours.

**Figure 3 cells-13-01942-f003:**
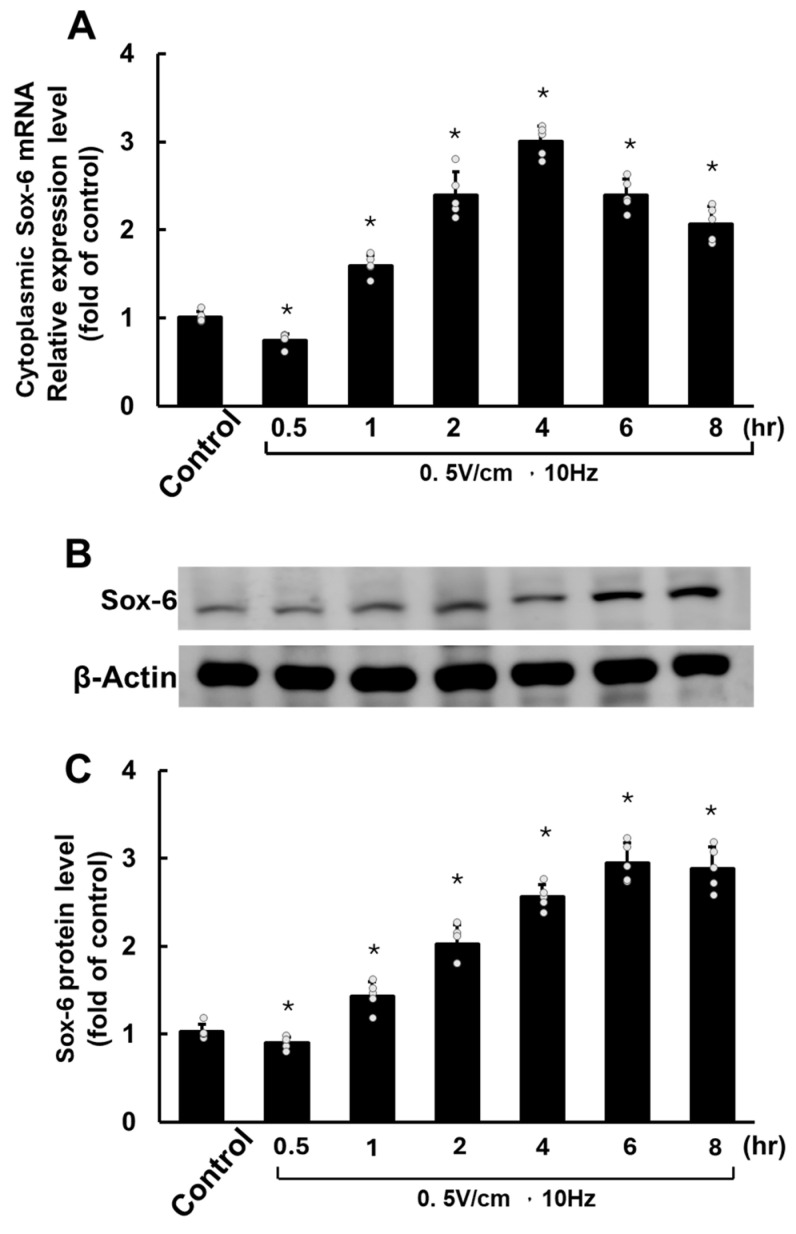
Rapid electrical stimulation (RES) impacts SOX6 mRNA and protein expression in HCF-aa. (**A**) Time course of SOX6 mRNA expression in HCF-aa subjected to RES (0.5 V/cm, 10 Hz) for 0.5 to 8 h. (**B**) Representative Western blot showing SOX6 protein expression in HCF-aa under RES (0.5 V/cm, 10 Hz) for 0.5 to 8 h. (**C**) Quantitative analysis of SOX6 protein expression corresponding to the Western blot in B. The control group was cultured for 8 h without RES for all panels. Data represent mean ± SEM from three independent experiments (n = 5 per group). * *p* < 0.001 vs. control group.

**Figure 4 cells-13-01942-f004:**
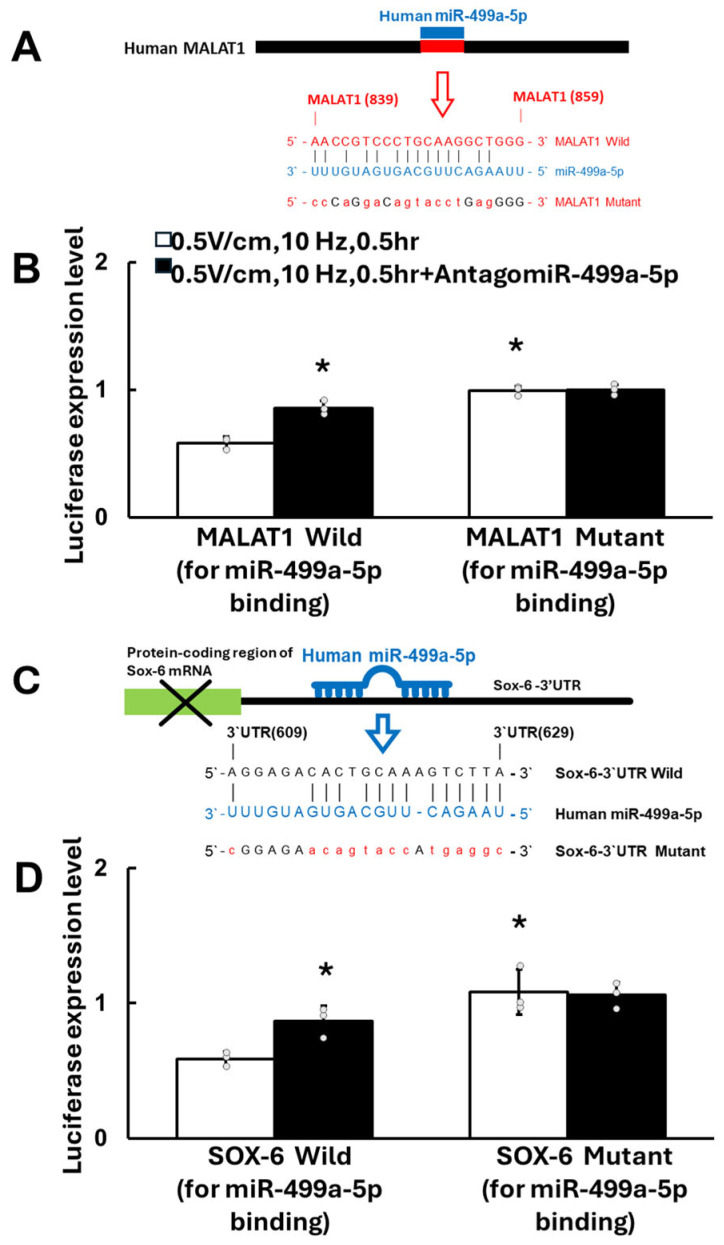
MALAT1 and SOX6 are target genes of miR-499a-5p in HCF-aa under rapid electrical stimulation (RES). (**A**) Putative binding site for miR-499a-5p on human MALAT1 transcript as predicted using luciferase assay. The schematic diagram shows that a human MALAT1 matching site (red rectangle) for miR-499a-5p (blue rectangle) was located from 839 bp to 859 bp of MALAT1 gene loci. (**B**) Luciferase activity assay of wild-type and mutant MALAT1-3′UTR in HCF-aa under RES (0.5 V/cm, 10 Hz, 0.5 h) with and without antagomir-499a-5p pretreatment. (**C**) Sequence of the SOX-6 3′UTR target site for miR-499a-5p binding (nucleotides 609–629). Human miR-499a-5p inhibits translation and/or negatively regulates Sox-6 mRNA stability by binding to the Sox-6 3′-untranslated region (3′-UTR), which causes Sox-6 expression to be suppressed. (**D**) Luciferase activity assay of wild-type and mutant SOX6-3′UTR in HCF-aa under RES (0.5 V/cm, 10 Hz, 0.5 h) with and without antagomir-499a-5p pretreatment. Pretreatment of antagomiR-499-5p enhanced expression of MALAT1 and SOX-6-3′ UTR luciferase activity in HCF-aa under RES. Data represent mean ± SEM from three independent experiments (n = 5 per group). * *p* < 0.05 vs. RES alone.

**Figure 5 cells-13-01942-f005:**
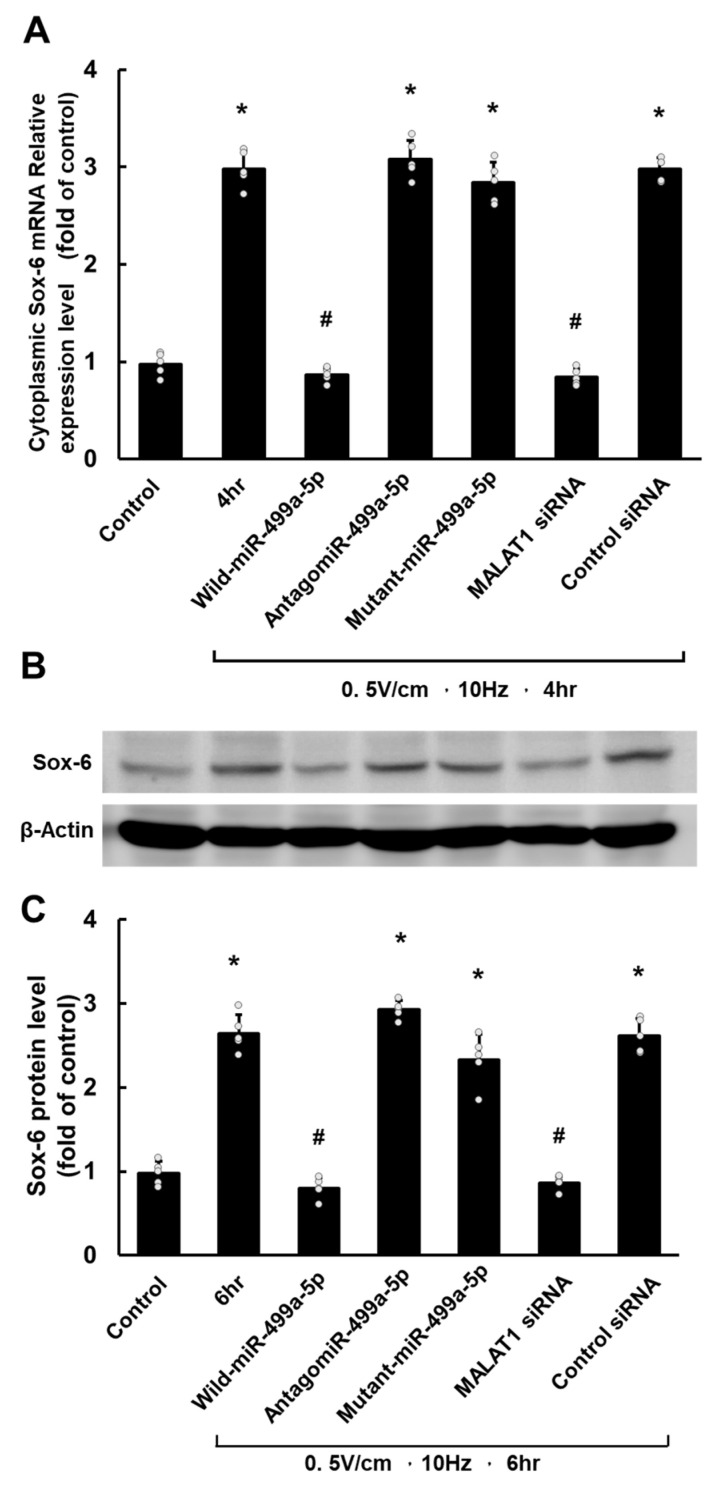
MALAT1 regulates SOX6 expression via miR-499a-5p in HCF-aa under rapid electrical stimulation (RES). (**A**) Quantitative analysis of SOX6 mRNA expression. (**B**) Representative Western blot showing SOX6 protein expression under various conditions. (**C**) Quantitative analysis of SOX6 protein expression. Conditions tested: Control (no RES), RES alone (0.5 V/cm, 10 Hz for 4 or 6 h), RES with wild-type miR-499a-5p overexpression, RES with antagomiR-499a-5p, RES with mutant miR-499a-5p, RES with MALAT1 siRNA, and RES with scramble siRNA. Data represent mean ± SEM from three independent experiments (n = 5 per group). * *p* < 0.05 vs. control, # *p* < 0.05 vs. RES alone.

**Figure 6 cells-13-01942-f006:**
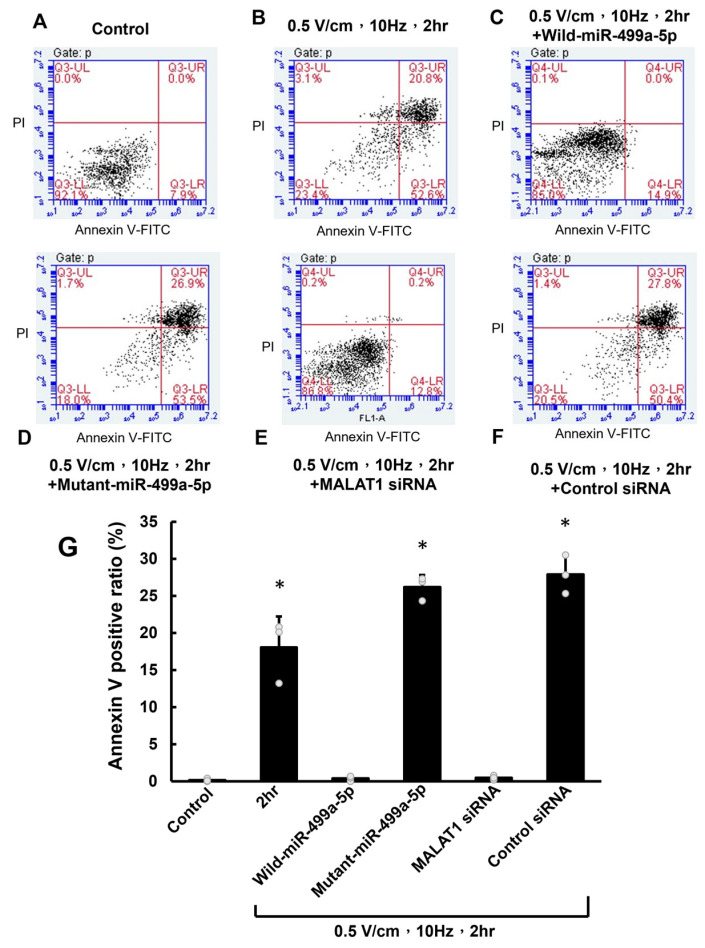
miR-499a-5p and MALAT1 mediate RES-induced apoptosis in HCF-aa. (**A**,**B**) Representative flow cytometry plots of propidium iodide–annexin V double-staining in control and RES-treated (0.5 V/cm, 10 Hz, 2 h) HCF-aa. (**C**) Flow cytometry plot showing the effect of miR-499a-5p overexpression on RES-induced apoptosis. (**D**) Flow cytometry plot demonstrating the impact of mutant miR-499a-5p on RES-induced apoptosis. (**E**) Flow cytometry plot showing the effect of MALAT1 siRNA on RES-induced apoptosis. (**F**) Flow cytometry plot demonstrating the impact of control siRNA on RES-induced apoptosis. (**G**) Quantitative analysis of apoptotic cells (positive for annexin V and propidium iodide) under all tested conditions. Data represent mean ± SEM from three independent experiments (n = 5 per group). * *p* < 0.05 vs. control.

**Figure 7 cells-13-01942-f007:**
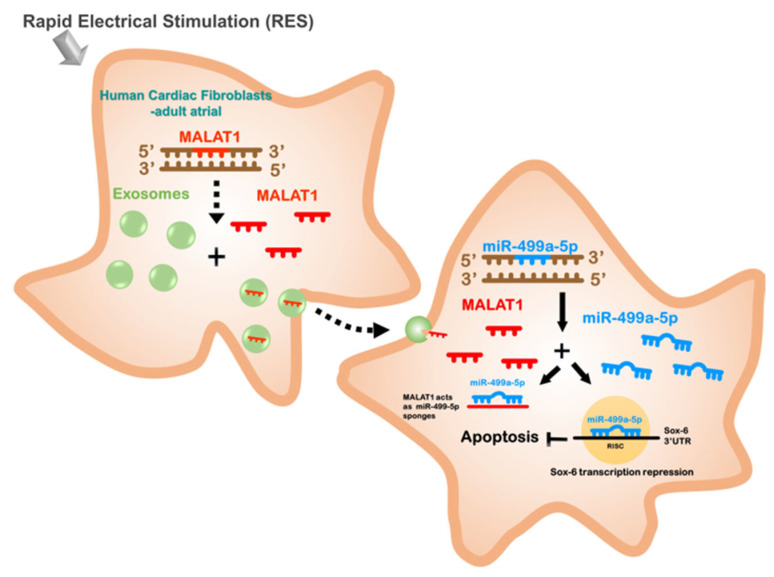
This schematic illustrates how rapid electrical stimulation (RES) enhances atrial fibrillation through the upregulation of exosomal MALAT1 in human cardiac fibroblasts from adult atria (HCF-aa). Increased exosomal MALAT1 leads to the suppression of miR-499a-5p, resulting in the upregulation of SOX6 expression in HCF-aa. This pathway highlights the critical role of HCF-aa-derived exosomal MALAT1 in the pathogenesis of atrial fibrillation under RES conditions. Consequently, HCF-aa-derived exosomal MALAT1 may serve as a potential therapeutic target for treating atrial fibrillation.

## Data Availability

The data presented in this study are available on request from the corresponding author.

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
