# Peer review of "Exosomal MALAT1 from Rapid Electrical Stimulation-Treated Atrial Fibroblasts Enhances Sox-6 Expression by Downregulating miR-499a-5p"

_cells, 2024, doi:10.3390/cells13231942_

Round 1

Reviewer 1 Report

Comments and Suggestions for Authors

General comments

Authors showed that rapid electrical stimulation (RES) increased exosomal MALAT1 (Metastasis-associated lung adenocarcinoma transcript 1) expression in 2 hours time. The rapid upregulation of exosomal MALAT1 was accompanied by miR-499a-5p levels. SOX6 mRNA and protein levels (a transcription factor critical in cardiac development and disease) increased, peaking at 4 and 6 hours respectively. Luciferase assays confirmed MALAT1 and SOX6 as miR-499a-5p targets. MALAT1 knockdown increased miR-499a-5p levels and reduced SOX6 expression. miR-499a-5p overexpression decreased SOX6 levels and inhibited RES-induced apoptosis. Authors suggest that RES induces the expression of MALAT1, which sequesters miR-499a-5p, leading to the upregulation of SOX6. They postulate that this regulatory axis plays a crucial role in the pathogenesis of atrial fibrilation by modulating fibroblast function and contributing to the development of fibrotic remodeling.

The study appears to have appropriate methodology. The data are clearly presented. The manuscript is well and clearly written.

There are few comments which may be useful:

Abstract

Line 15 – please expand the abbreviation MALAT1

Line 17 - please expand the abbreviation SOX6

Materials and Methods

Line 93  – figure should be written with a capital letter

Line 182 – what kind of protein assay was used ?

Author Response

Response to Reviewer 1

Abstract

Line 15 – please expand the abbreviation MALAT1

Line 17 - please expand the abbreviation SOX6

Materials and Methods

Line 93  – figure should be written with a capital letter

Line 182 – what kind of protein assay was used ?

Ans: Thank you for your correction. We have made the requested changes as follows:

Abstract, Line 15: Expanded MALAT1 to Metastasis Associated Lung Adenocarcinoma Transcript 1.

Abstract, Line 17: Expanded SOX6 to SRY-box Transcription Factor 6

Materials and Methods, Line 93: Updated "figure" to "Figure" with a capital letter.

Materials and Methods, Line 182: Specified that we used Bio-Rad Protein Assay to measure protein content.

Reviewer 2 Report

Comments and Suggestions for Authors

The article coauthored by Chuang et al. presents a comprehensive investigation into the impact of rapid electrical stimulation on the expression levels of MALAT1, miR-499a-5p, and SOX6 in human atrial fibroblasts (HCF-aa). The results are highly intriguing and provide novel insights into the molecular mechanisms that may contribute to the pathogenesis of atrial fibrillation. At the outset, I would like to commend the authors on their choice of research topic, which is currently of great relevance.

The research methodology was meticulously planned and involved a range of techniques, including cell culture, precise electrical stimulation settings, and a variety of molecular analyses. The study employed both quantitative polymerase chain reaction (PCR) and protein analyses (Western blot) to accurately monitor changes in the expression levels of selected genes and proteins. The utilization of luciferase as a verification tool for the relationship between miR-499a-5p, MALAT1, and SOX6 enabled the unequivocal confirmation of the role of these molecules in the mechanism under investigation. The utilization of antagomiR-499a-5p and siRNA for MALAT1 further enhanced the precision of the assay, enabling the accurate tracing of the relationship between the various components of the regulatory pathway under analysis.

The selection of stimulation parameters (frequency 10 Hz, voltage 0.5V/cm) was informed by prior studies, reflecting a meticulous and evidence-based approach to the research project. Furthermore, apoptosis analysis by flow cytometry yielded crucial insights into the impact of miR-499a-5p and MALAT1 regulation on the survival of RES-treated fibroblasts.

The results are presented in a highly detailed and comprehensive manner. The accompanying illustrations are lucid and accurately delineated.

The article is based on a robust and current literature base, which lends credibility and context to the research conducted. The sources used are drawn from high-quality scientific journals, and their selection demonstrates a comprehensive and up-to-date review of the literature.

In my opinion, this article should be accepted in current form.

Author Response

Thank you for your positive and encouraging feedback on our manuscript. We greatly appreciate your recognition of our work’s contribution to understanding the molecular mechanisms of atrial fibrillation. Thank you once again for your thoughtful review.
